# Effect of Particle Size on the Mechanical Properties of TiO_2_–Epoxy Nanocomposites

**DOI:** 10.3390/ma14112866

**Published:** 2021-05-27

**Authors:** Young-Min Choi, Seon-Ae Hwangbo, Tae Geol Lee, Young-Bog Ham

**Affiliations:** 1Department of Thermal Systems, Korea Institute of Machinery and Materials, Daejeon 34103, Korea; anaud007@kimm.re.kr; 2Nanosafety Team, Safety Measurement Institute, Korea Research Institute of Standards and Science, Daejeon 34113, Korea; hbsa@kriss.re.kr (S.-A.H.); tglee@kriss.re.kr (T.G.L.)

**Keywords:** polymer nanocomposites, dispersoid, nano-TiO_2_ suspensions, ultrasonic dispersion, focused ultrasonication

## Abstract

This study investigated the effects of the packing density and particle size distribution of TiO_2_ nanoparticles on the mechanical properties of TiO_2_–epoxy nanocomposites (NCs). The uniform dispersion and good interfacial bonding of TiO_2_ in the epoxy resin resulted in improved mechanical properties with the addition of nanoparticles. Reinforcement nano-TiO_2_ particles dispersed in deionized water produced by three different ultrasonic dispersion methods were used; the ultrasonication effects were then compared. The nano-TiO_2_ suspension was added at 0.5–5.0 wt.%, and the mechanical and thermal properties of TiO_2_–epoxy NCs were compared using a universal testing machine, scanning electron microscopy (SEM), Fourier-transform infrared spectroscopy (FT-IR), and differential scanning calorimetry (DSC). The tensile strength of the NCs was improved by the dispersion strengthening effect of the TiO_2_ nanoparticles, and focused sonication improved the tensile strength the most when nano-TiO_2_ suspensions with a particle size of 100 nm or smaller were used. Thus, the reinforcing effect of TiO_2_ nanoparticles on the epoxy resin was observed, and the nano-TiO_2_ suspension produced by focused sonication showed a more distinct reinforcing effect.

## 1. Introduction

Epoxy resin (EP) is an important industrial material that has many applications in the electronics, automobiles, and aerospace fields, owing to its high strength and stiffness, resistance to chemicals, low contraction during curing, excellent corrosion resistance, and thermal characteristics [1,2,3]. However, its other characteristics such as brittleness, poor resistance to crack propagation, and low wear resistance limit its applications [4,5,6]. To this end, several studies have investigated various fillers that can be added to improve the properties of the matrix.

Composite technology and nanotechnology are being extensively researched to solve these chronic problems that cannot be solved using single-layer materials. Nanocomposite technology has gained considerable research interest in all disciplines including engineering, physics, chemistry, and medicine; furthermore, it is believed to promote technical development in various fields such as information, electronics, materials, and energy [7]. Nanocomposites (NCs) reinforced with nanosized materials (1–100 nm size distribution) can be applied to a variety of materials including polymer-based NCs. Polymer nanocomposites (PNCs) are produced by combining lightweight, flexible polymers with low production costs as a matrix with inorganic nanoparticles that exhibits excellent mechanical and electromagnetic properties [8,9,10]. A PNC is a polymer matrix modified with reinforcement nanoparticles at a low ratio (<5.0 wt.%). Various nanoparticles, such as carbon nanotubes, titania (TiO_2_), silica, and alumina are used as the reinforcement particles [11,12,13,14]. These PNCs exhibit excellent mechanical, physical, and chemical properties; however, these properties strongly depend on the interfacial bond strength between the polymer matrix and reinforcement particles. The cohesion of nanoparticles and the high viscosity of polymers makes it considerably difficult to uniformly disperse nanoparticles of 5 wt.% or lower. Therefore, a mechanism that can uniformly disperse nanoparticles in a highly viscous liquid is highly desirable in academic circles [15,16,17].

The performance of nanoparticle-reinforced PNCs as a function of dispersion and interfacial interaction has been investigated in many studies; the results have suggested that good dispersion and strong interfacial interaction can effectively transfer the stress from the polymer matrix to the nanomaterials [18,19]. Various dispersion methods such as extrusion, ball milling, three-roll milling, solvent casting, and functionalization of nanoparticles have been employed. Among these methods, the mechanical mixing method is most commonly applied to create violent mixing, thereby generating turbulence on a large scale to nanoparticles. However, this method cannot prevent nanoparticle aggregation into the polymer matrix [20,21,22]. Therefore, ultrasonic dispersion methods are used instead of mechanical mixing methods to achieve high homogeneity [23,24]. Ultrasonic waves are classified into low- and high-frequency waves; the latter can decompose aggregates and separate particles accumulated on the microchannel surface by generating cavitation microbubbles. In contrast, the former form an ultrasonic sound field and lead to acoustic radiation and streaming effects, which facilitates biological and chemical treatments such as cell separation and fluid mixing [25,26,27,28,29]. Ultrasonic cavitation is an effective method for dispersing nanoparticles. As compression and rarefaction cycles are switched sequentially, high-intensity ultrasonic waves spread to the liquid polymer matrix; simultaneously, microbubbles of vacuum are created and collapsed in the matrix. When these microbubbles collapse, the local temperature changes to 10^5^ Ks^−1^ because of the shockwave, and a pressure of several megapascals is generated. This energy can be used to disperse nanoparticles [30,31].

This study compared the dispersion of nano-TiO_2_ suspensions using three types of ultrasonic systems (bath sonication, probe sonication, and the novel focused sonication method we developed and reported in a previous paper [32]) to verify the nanoparticle dispersion efficiency of a proprietary focused sonication system. In detail, focused ultrasound uses cylindrical piezoelectric ceramics to dissolve agglomeration through the energy collected in the center. Generally, the surface charge of particles that prevent agglomeration is improved, to have a zeta potential value of about 40–50 mV in the dispersion process, which ensures long-term dispersion stability that does not re-aggregate for more than 1 year. We had confirmed that even when lyophilized particles were re-dispersed in water, they were dispersed to less than 100 nm without aggregation. Thus, the NCs based on EP were produced using the nano-TiO_2_ suspensions obtained by each ultrasonication method in this study; then, the physical properties of the TiO_2_–epoxy NCs were compared based on the added amount.

## 2. Experiment

### 2.1. Materials

TiO_2_ nanoparticles used in this study were type P25 (CAS: 13463-67-7) purchased from EVONIK Co. Ltd. (Essen, Germany); they had an average particle size of 25 nm and a density of 3.78 g/cm^3^. Figure 1 shows the shape of the TiO_2_ nanoparticles observed using transmission electron microscopy (TEM) (JEM-ARM200F, JEOL, Tokyo, Japan).

The polymer matrix was produced by mixing an EP and hardener supplied by Kukdo Chemical Co. Ltd. (Seoul, Korea). YD-128 (CAS: 25068-38-6) of the bisphenol-A type was used as the EP, and KBH-1085 (CAS: 25134-21-8) of nadic methyl anhydride (NMA) was used as the hardener. An accelerator (CAS: 103-83-3) of benzyl dimethyl amine (BDMA) was used as the hardening accelerator and was supplied by NEW SEOUL CHEMICAL (Seoul, Korea). YD-128, KBH-1085, and the accelerator were mixed at a manufacturer-recommended weight ratio of 100:60:2.5. Figure 2 shows the molecular structures of the EP, hardener, and hardening accelerator.

### 2.2. Sample Preparation

In this study, the PNC was produced by dispersing TiO_2_ nanoparticles (1.0 wt.%) in 100 mL of the deionized water and by adding it into a matrix. The nano-TiO_2_ suspension was dispersed using bath-type, probe-type, or focused-type sonication. SD-300H (Sungdong Ultrasonic Co., Seoul, Korea) and Branson SFX 550 (Emerson Co. Ltd., Saint Louis, MO, USA) were used for bath- and probe-type sonication, respectively. The results were compared with those of a proprietary focused sonication system [32].

Delivered sonic energy (DSE) (J/mL) is an important parameter in ultrasonication; it is the actual ultrasonic energy delivered to the sample. The mean particle size of TiO_2_ when DSE was applied to the same volume is compared in this study [33]. The DSE was calculated using
P = (dT/dt)MC_p_,(1)
DSE = P × (t/V),(2)
where P, dT/dt, M, C_p_, t, and V denote the ultrasonic energy obtained by the calorimetric method [J/s], variation in temperature at time t [K/s], mass of the solution [g], heat capacity of water, time [s], and volume of the solution (mL), respectively.

The DSEs obtained using the three types of sonication methods are summarized in Table 1. To verify the particle size distribution of the TiO_2_ suspension dispersed at different ultrasonic energies, the mean particle size was measured after ultrasonication using dynamic light scattering (Zetasizer Nano ZS, Malvern Panalytical Ltd., Malvern, UK).

The particle size of TiO_2_ was 25 nm; however, it was agglomerated to approximately 500 nm [34]. After dispersion with the three sonication methods, the mean sizes of the nanoparticles were 96.4 nm, 155.3 nm, and 206.8 nm, as shown in Figure 3. Nano-TiO_2_ suspensions with a diameter of 100 nm were produced when the particles were dispersed by focused sonication. Furthermore, the intensity of the size distribution was higher than that of the others, which indicated that the particle sizes were more uniform because of the focused sonication.

Figure 4 shows an overview of the PNC manufacturing process. In this study, NCs with four different compositions were manufactured by adding nano-TiO_2_ suspensions (0.5 wt.%, 1.0 wt.%, 3.0 wt.%, and 5.0 wt.%) to an EP. It was difficult to disperse the EP with nanoparticles because of its high viscosity; therefore, the EP and TiO_2_ suspensions were dispersed using a stirring device and probe-type sonication. After mixing for 30 min at 700 rpm at 293.15 K using a mechanical agitator OS20-Pro (Korea Process Technology Co., Ltd., Seoul, Korea), they were dispersed for 180 min at 293.15 K using an ultrasonic dispersion system VC-505 (Sonics & Materials Inc. (Newtown, CT, USA). The residual gas was removed from the EP with a TiO_2_ suspension dispersed through a degassing process for 180 min in a 333.15 K vacuum oven. Then, the hardener and hardening accelerator were added and mixed for 5 min by hand stirring to blend and evenly distribute it with minimal bubbles; an additional degassing process was applied for 30 min in a 333.15 K vacuum oven. The TiO_2_–epoxy NCs were fabricated as tensile specimens (shape and dimensions as shown in Figure 5) hardened in two steps of 398.15 K for 60 min and 423.15 K for 180 min (Figure 4).

### 2.3. Characteristics of the Nanocomposites

The mechanical properties of 12 different batches of TiO_2_–epoxy NCs (namely, four different compositions and three different sonication methods) were measured using tensile tests and compared with those of pure epoxy resin samples. As shown in Figure 5, eight specimens per batch were molded into dog-bone-shaped tensile test specimens, which were tested in the tension mode of a universal testing machine M350-10CT (Testometric Co. Ltd., Rochdale, UK) at a crosshead speed of 2 mm/min. Then, the five median values were determined from each batch of eight samples, and the average of these five median values was calculated. In addition, the fracture surface was analyzed using SEM (SEM; SU5000, Hitachi, Tokyo, Japan) after the tensile test.

The thermal properties of one sample per batch were examined using DSC, which can analyze thermal properties such as glass transition temperature, crystallization temperature, and melting temperature using the calorie difference based on the temperature gradient between the reference material and the sample. Quantitative data can be obtained from the positions, shapes, and areas of the peaks in a DSC curve. In this study, the thermal properties of pure epoxy and TiO_2_–epoxy NCs were analyzed using the Perkin Elmer Diamond model (Perkin Elmer, Waltham, MA, USA) at a heating rate of 10 K/min from 273.15 K to 673.15 K in a nitrogen (N_2_) atmosphere.

Attenuated total reflectance Fourier-transform infrared (ATR-FTIR) spectroscopy was also used on one sample per batch to verify the bonding structure of the samples. ATR-FTIR spectroscopy is useful for measuring the surfaces of polymers such as sponges and adhesives, which are difficult to process. The IR beam from the interferometer is transmitted into a diamond crystal that results in an internal reflection at the boundary with the sample, because it has a greater refractive index than the sample in contact on both sides. Meanwhile, the IR spectrum that reflects the vibration and rotational motions of molecules are measured, and a qualitative analysis of the sample including molecular species is performed. The pure epoxy and TiO_2_–epoxy NCs were analyzed using ALPHA-P (Bruker, Billerica, MA, USA); the ATR-FTIR spectra were collected by scanning the sample surface in the range of 400–4000 cm^−1^.

## 3. Results and Discussion

### 3.1. Mechanical Properties of Nanocomposites

Dog-bone-shaped specimens were fabricated, and the tensile strengths were measured to examine the changes in the mechanical properties of the TiO_2_–epoxy NCs. Figure 6 shows the average tensile strengths of the NCs after sonication of the nano-TiO_2_ suspension. The average tensile strength of the composites with the nano-TiO_2_ suspension added was up to 15% higher than that of the pure EP. In particular, the tensile strengths of the NCs with 1.0 wt.% nano-TiO_2_ suspension processed by focused sonication was higher (92.9 MPa) compared to those processed using probe sonication (91.1 MPa) or bath sonication (89.7 MPa). Usually, bath sonication can handle large amounts at once, although it has been known to be less efficient at dispersion, and probe sonication has a strong irradiation energy, but it also has blind spots such as on the upper side of the probe. On the other hand, compared with the former sonication methods, it was determined that focused sonication had a positive effect on the TiO_2_ dispersion in the EP matrix. Due to the driving characteristics of focused sonication, it could continuously apply ultrasonic energy to the circulating object [32].

The tensile strength of the composites tended to decrease gradually when the added amount of nano-TiO_2_ suspension that passed through the ultrasonic dispersion system was 3.0 wt.% or higher. The decreased tensile strength of the NCs reinforced with nano-TiO_2_ suspension above a certain amount seems to be attributed to the following two problems: (1) The deionized water among the component materials of the nano-TiO_2_ suspension fabricated in this study had an adverse effect on the polymerization process of the EPs with hydroxyl and carboxyl groups. Therefore, the residual water in the TiO_2_–epoxy NCs needed to be checked via thermal analysis and the improvements should be identified [35,36]; (2) The interference of the epoxy polymerization and the reduced effect of particle reinforcement was attributed to the agglomeration of TiO_2_ nanoparticles. Adding a large quantity of nanoparticles into a highly viscous EP causes agglomeration because of the attraction between particles. Accordingly, the agglomerated nanoparticles serve as impurities in the resin; therefore, they decrease the mechanical properties [37,38,39]. In this study, when PNCs were produced by adding a nano-TiO_2_ suspension produced by the same sonication method in the range of 0.5–5.0 wt.%, the addition of 1.0 wt.% was found to be the optimal amount.

### 3.2. Microstructural and Morphological Analysis

Microstructures at 200× magnification were observed using SEM to examine the failure behavior of the TiO_2_–epoxy NCs. Figure 7 shows the fracture surface images after the tensile testing of pure EP and epoxy NCs reinforced with TiO_2_ using different sonication methods. Pure EP was very clear: from Figure 7a, the fracture surface of neat epoxy showed a very smooth pattern where cracks propagated arbitrarily without restrain, indicating the brittle fracture of neat epoxy without resistance to crack initiation and transmission. Compared to the fracture surface of the pure EP, the fracture surface of NCs with TiO_2_ nanoparticles added in Figure 7b–d showed a rough cliff-like pattern (blue arrow: flat cross-section, and red arrow: rough cross-section). In general, more fracture energy is dissipated for greater roughness. These observations reveal the generation of a rough surface by the deflection of propagating crack fronts on the loading of TiO_2_ in the EP matrix. In other words, the cracks progressed relatively rapidly without high resistance at the fracture surface of the pure epoxy specimen; furthermore, the crack resistance of the TiO_2_–epoxy NCs improved with the addition of nanoparticles, which acted as a factor that increased the tensile strength [40,41].

Figure 8 shows the SEM images with 500× magnification for the fracture surfaces of the nanocomposite reinforced with nano-TiO_2_ suspension dispersed by focused sonication. The characteristic brittleness and low fracture toughness of the pure epoxy are a product of their high cross-link densities, which resulted in the poor absorption of energy during the fracture. These factors frequently lead to mirror-like fracture surfaces, as shown in Figure 8a. Changes in the fracture surface caused by the added nanoparticles were examined. Compared to that for the pure epoxy, the fracture surface of NCs contained greater roughness, as shown in Figure 8b–e. The coarse multiplane features on the TiO_2_–epoxy NCs fracture surface suggest that the TiO_2_ nanoparticles induced the deflection of propagating crack fronts. Furthermore, NCs with 3.0 wt.% and 5.0 wt.% TiO_2_ showed a decreased roughness compared to the NCs with 1.0 wt.% nano-TiO_2_ suspension. The PNCs are effective only if the nanoparticles are well dispersed in the surrounding polymer matrix. In addition, the agglomeration of nanoparticles should not be overlooked, because nanoparticles provide a very specific surface area in the polymer matrix [10]. In other words, over-addition of nanoparticles causes the agglomeration of particles, which reduces the particle reinforcement effect. Consequently, the tensile strength could be decreased, as shown in Figure 6 above.

### 3.3. FT-IR Spectroscopy

In this study, PNCs were produced using nano-TiO_2_ suspensions processed by different sonication methods but with the same content, and their physical properties were compared. As listed in Table 1, the particle size varied according to the sonication method used. The focused sonication showed an average particle size of 96.4 nm, which is smaller than those of other sonication methods. In general, crosslinking density and hydrogen bonding density are significantly affected by the content and size of the nanoparticles and the packing density in the polymer matrix. Therefore, singularities based on the bonding type and added amount of TiO_2_ in the polymer nanocomposite network were examined by FT-IR analysis.

Figure 9 shows the FT-IR spectra of pure epoxy and TiO_2_–epoxy NCs in the range of 400–4000 cm^−1^. The O–H bond could appear at 3500–3000 cm^−1^, and H–O–H bonds are normally detected at 1700–1200 cm^−1^. These bonds could occur in the presence of water, and the results of FT-IR analysis are shown in this study. Therefore, it is assumed that there was no residual water. The peaks at 2850–2950 cm^−1^ indicated the C–H group, whereas those at 1730–1740 cm^−1^ and 1510 cm^−1^ indicated the C=O and aromatic groups, respectively. The TiO_2_–epoxy NCs processed by focused sonication showed peak intensities similar to those of pure epoxy. Overall, the FT-IR patterns of all samples including the pure EP were almost the same, which suggests that there was no chemical bonding between TiO_2_ and the EP.

### 3.4. Thermal Property of Nanocomposites

Nanoparticles play a key role in improving the thermodynamic properties of EPs. Among them, TiO_2_ is widely used in industries because of its diverse functions and synthesis methods. The state of materials can be analyzed using DSC based on temperature changes with high speed and precision, and it is used in research investigating the physical properties of polymers. In particular, DSC is widely used for examining polymeric materials to determine their thermal transitions. Important thermal transitions include the glass transition temperature (T_g_), crystallization temperature (T_c_), and melting temperature (T_m_). The epoxy resin of bisphenol-A-type, an amorphous polymer, was used in this study; thus, we expected to obtain acceptable T_g_ and T_m_ values. The T_g_ is a very important thermal property in polymeric systems. The unit of polymers is a certain segment in the molecular chain, and the thermal motion by the rotation of this segment is called micro-Brownian motion. The temperature at which micro-Brownian motion starts is called the glass transition temperature, and it is also referred to as the second transition temperature, which is a unique property of polymers.

Figure 10 shows the DSC curves of the pure epoxy and TiO_2_–epoxy NCs. A glass transition point was observed at 350–450 K, wherein the endothermic reaction of pure EP appeared distinctly. The glass transition temperature of the pure EP is 355 K. Furthermore, the glass transition temperatures of the NCs processed by focused sonication, probe sonication, and bath sonication were 400.0 K, 375.7 K, and 360.0 K, respectively. This confirmed that the thermal properties of the TiO_2_–epoxy NCs were improved compared to those of the pure EP [42,43]. The NCs in which the nano-TiO_2_ suspensions were dispersed by focused sonication showed the best thermal properties, and therefore, thermal stability could be expected.

On the other hand, there were no other reactions than phase transitions of the nanocomposites. We fabricated the nanocomposites using an aqueous solution-based TiO_2_ suspension in this study, and we have confirmed that the residual moisture could have an adverse effect on the nanocomposites. However, compared to pure epoxy resin, the specificity peak of nanocomposites was not shown in the DSC analysis.

## 4. Conclusions

In this study, the effect of TiO_2_–epoxy size on the mechanical properties of particle NCs was investigated. In addition, nano-TiO_2_ suspensions were dispersed separately using the existing bath sonication and probe sonication methods to examine the dispersion efficiency of a proprietary focused sonication dispersion system. A nano-TiO_2_ suspension with a mean particle size of 96.4 nm was obtained using the focused sonication dispersion system. TiO_2_ particle-reinforced epoxy NCs were fabricated, and their mechanical properties were compared. The following conclusions could be drawn based on the results.

The tensile strength of the TiO_2_–epoxy NCs was improved compared to that of the pure EP. The highest tensile strength was observed when 1.0 wt.% of TiO_2_ suspension (mean particle size 100 nm or lower) dispersed by focused sonication was added. However, the tensile strength of more than 3.0 wt.% TiO_2_-reinforced epoxy NCs decreased regardless of particle size;The SEM analysis of the fracture surface indicated that the crack resistance of the TiO_2_–epoxy NCs was greater than that of the pure EP; this influences the tensile strength;The FT-IR spectra of the pure EP and TiO_2_–epoxy NCs were almost the same, which suggests that chemical bonding with TiO_2_ did not occur. The TiO_2_–epoxy NCs showed improved thermal properties compared to pure EP;The glass transition temperature of the TiO_2_-reinforced epoxy NCs with a particle size of 100 nm or less (dispersed by focused sonication) was the highest at 400 K; this implies that the performance improvement of the nanocomposite was affected by the size of the nanoparticles;Both the tensile strength and thermal properties improved when a suspension with a particle size of 100 nm or less was used; this was confirmed to be greatly influenced by the nanoparticle dispersion method.

Continuous research is needed to further optimize and standardize the focused ultrasound method used in this study so that it can be applied to various NCs.

## Figures and Tables

**Figure 1 materials-14-02866-f001:**
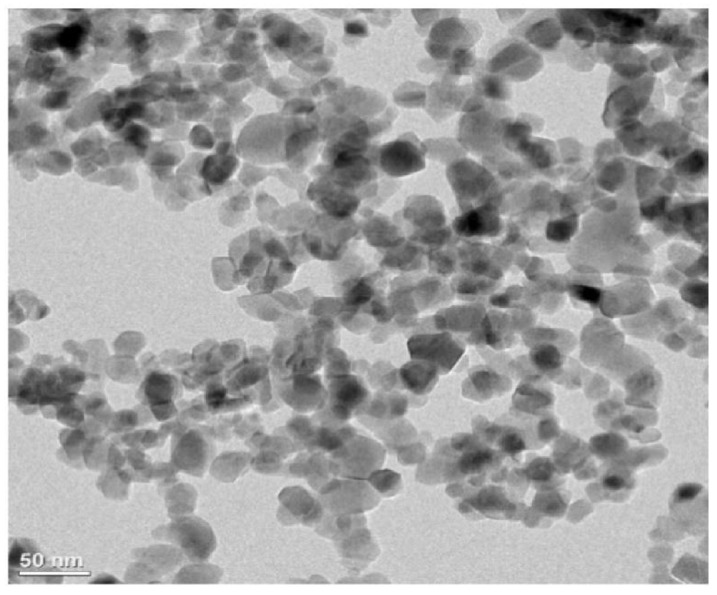
TEM image of the P25 purchased from EVONIK Co., Ltd.

**Figure 2 materials-14-02866-f002:**
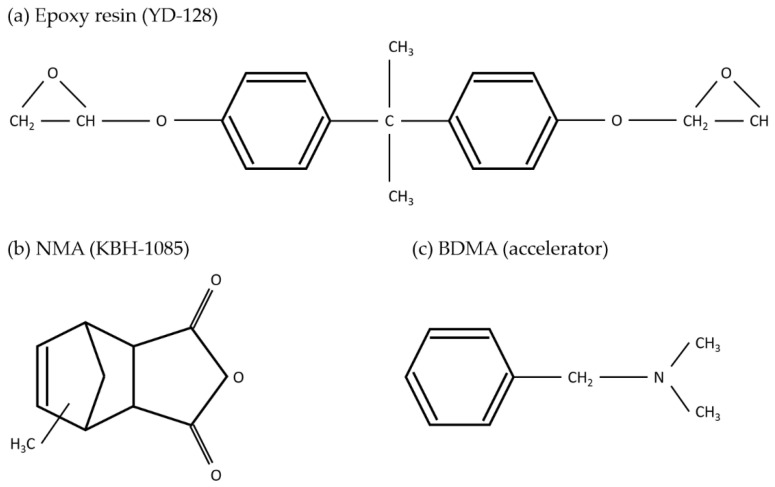
Chemical structures of (**a**) epoxy resin, (**b**) hardener, and (**c**) hardening accelerator.

**Figure 3 materials-14-02866-f003:**
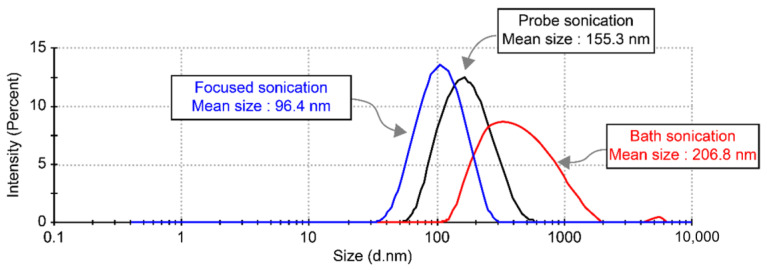
Size distribution of TiO_2_ nanoparticles in deionized water based on sonication type.

**Figure 4 materials-14-02866-f004:**
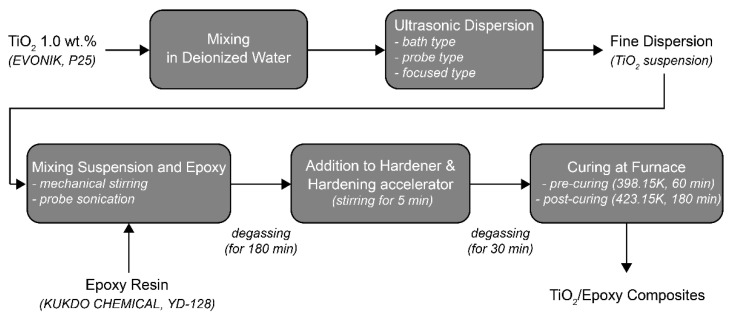
Flowchart of TiO_2_–epoxy nanocomposite preparation.

**Figure 5 materials-14-02866-f005:**
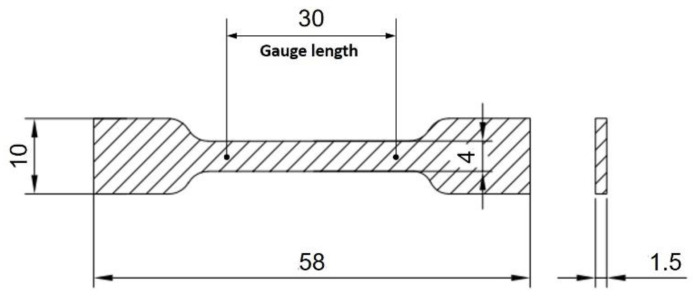
Dimensions for the tensile test specimen.

**Figure 6 materials-14-02866-f006:**
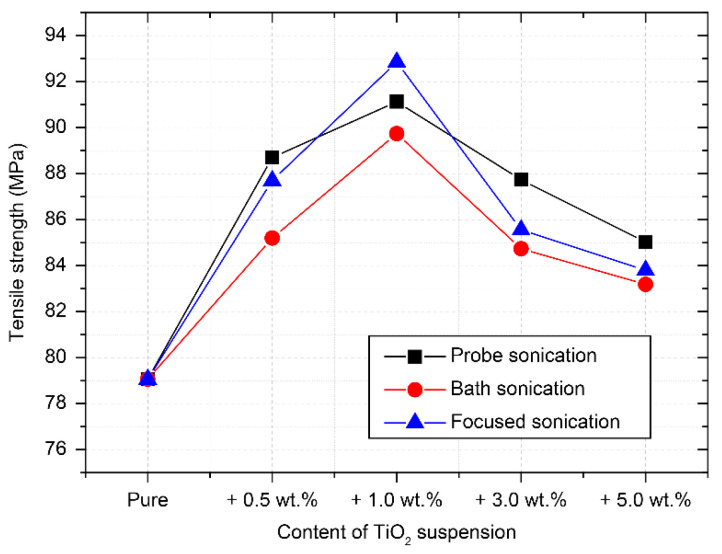
Tensile strength of the TiO_2_–epoxy nanocomposites in accordance with the sonication method.

**Figure 7 materials-14-02866-f007:**
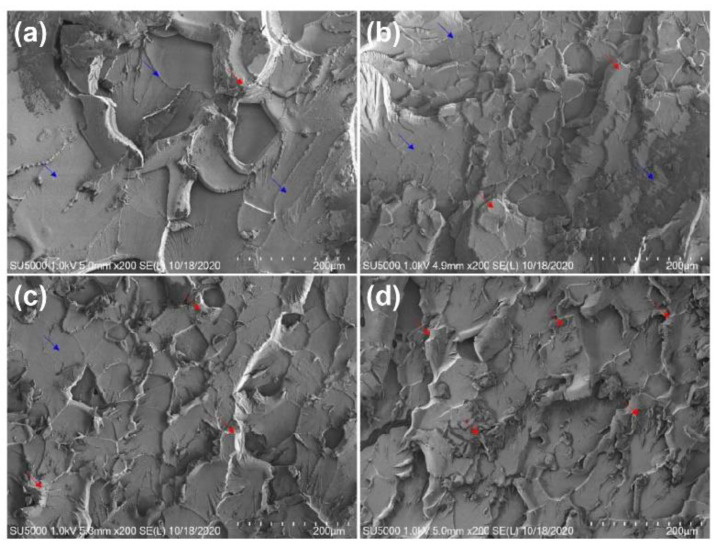
SEM images of (**a**) the pure epoxy resin and the 1.0 wt.% TiO_2_–epoxy nanocomposites with (**b**) bath sonication, (**c**) probe sonication, and (**d**) focused sonication.

**Figure 8 materials-14-02866-f008:**
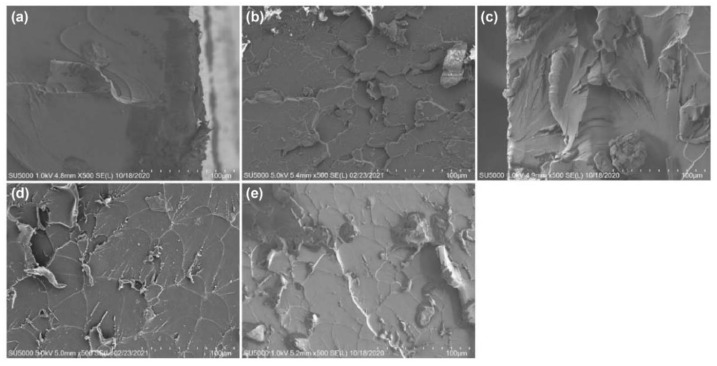
SEM images of (**a**) pure epoxy resin, (**b**) 0.5 wt.%, (**c**) 1.0 wt.%, (**d**) 3.0 wt.%, and (**e**) 5.0 wt.% TiO_2_–epoxy nanocomposites with focused sonication.

**Figure 9 materials-14-02866-f009:**
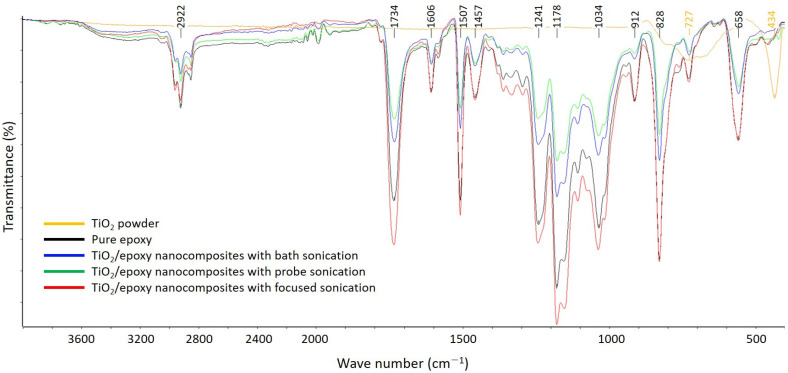
FT-IR spectra of the TiO_2_ powder, pure epoxy resin, and the 1.0 wt.% TiO_2_–epoxy nanocomposites.

**Figure 10 materials-14-02866-f010:**
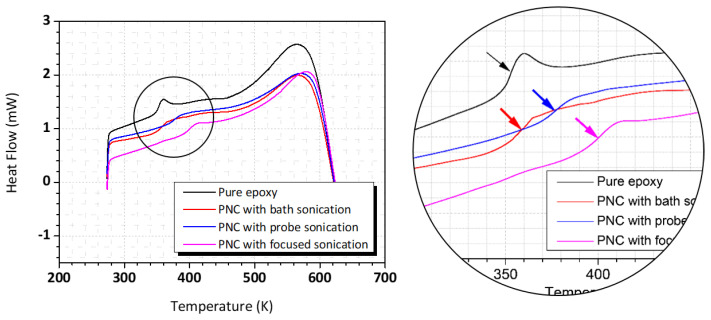
DSC curve of the 1.0 wt.% TiO_2_–epoxy nanocomposites in accordance with the sonication method conducted within the range of 200–700 K.

**Table 1 materials-14-02866-t001:** Specifications of the three types of sonication methods.

Specification	Bath Sonication	Probe Sonication	Focused Sonication
Frequency	40 kHz	20 kHz	400 kHz
Solution volume	100 mL	100 mL	100 mL
Irradiation time	32 h 12 min	1 h 47 min	2 h
Ultrasonic energy (P)	1.7 J/s	29.6 J/s	27.0 J/s
Delivered sonic energy (DSE)	194.7 J/mL	194.7 J/mL	194.7 J/mL
Median particle size	206.8 nm	155.3 nm	96.4 nm
Range of particle size distribution	122–5560 nm	60–531 nm	37.8–255 nm

## Data Availability

Data available in a publicly accessible repository.

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
