# Peer review of "Effect of Particle Size on the Mechanical Properties of TiO2–Epoxy Nanocomposites"

_materials, 2021, doi:10.3390/ma14112866_

Round 1
Reviewer 1 Report
I have evaluated the manuscript entitled: "Effect of Particle Size on the Mechanical Properties of TiO2–Epoxy Nanocomposites" (Manuscript ID: materials-1185292). This study describes the effects of the packing density and particle size distribution of TiO2 nanoparticles on the mechanical properties of TiO2–epoxy nanocomposites. Three different dispersion methods (based on ultrasonic dispersion) are presented to obtain homogeneous organic/inorganic nanocomposite samples.
1) In general, I did not find great originality/novelty in this work. It is very similar to several other works that have been already published. (Example https://doi.org/10.1016/j.ultsonch.2010.03.011) I suggest to stress more the attention of the reader on the proposed innovation. Why this study is important? What is the potential improvement that could be obtained from this work?
2) Does the epoxy resin solubilizes in water when you mix it with TiO2 aqueous dispersion? i expect that it does not. how could you remove the excess of water?
3) Moreover, the article merely summarises the obtained results without any explanation or real discussion. I suggest to expand the discussion part.
4) I suggest to briefly report in the introduction the in situ approach as dispersion method. I suggest to consider the following works:
- Phys. A 122 (2016) 1075; DOI:10.1007/s00339-016-0606-6
- Macromolecular Material Engineering 305 (2020) 2000017; DOI:1002/mame.202000017
Author Response
Dear reviewer,
First of all, thank you for your valuable comments concerning our manuscript entitled " Effect of Particle Size on the Mechanical Properties of TiO2–Epoxy Nanocomposites " by Young Min Choi, Seon Ae Hwangbo, Tae Geol Lee and Young Bog Ham* for publication in Micromachines journal.
Your feedback and comments are all valuable and very helpful for revising and improving our paper. We have studied comments carefully and have made detailed and cautious English language corrections in addition to improving the quality of graphics used in manuscript and we hope that the revised manuscript meets your publication requirements.
Please check the attached file.
Kind regards,
Dr. Young Min Choi, Dr. Seon Ae Hwangbo, Dr. Tae Geol Lee and Dr. Young Bog Ham

Reviewer 2 Report
In this paper Authors investigated the effects of the packing density and particle size distribution of TiO2 nanoparticles on the mechanical properties of TiO2–epoxy nanocomposites. Authors used reinforcement nano-TiO2 particles dispersed in deionized water produced by three different ultrasonic dispersion methods. In next step the ultrasonication effect was compared. Authors, Authors also compared mechanical and thermal properties of TiO2-epoxy nanocomposites. For this purpose, they used a universal testing machine, scanning electron microscopy (SEM), Fourier-transform infrared spectroscopy (FT-IR) and differential scanning calorimetry (DSC).
The originality the concepts, the significance and the methods are good. The completeness and the organization of manuscript of the paper are good. The organization of the manuscript is good. In my opinion the technical treatment is plausible and free of technical errors. The paper is interesting, written in a transparent and legible way.
Below I presented some remarks that came to my mind during reading:
- Lines 82-94: Can the authors provide CAS numbers for the used substances?
- Lines 135-137: On what basis were nanocomposite concentrations selected? This information should be provided.
Author Response

(The authors gave the same response as above.)

Reviewer 3 Report
The manuscript investigated the effect of particle size on the mechanical properties of TiO2–epoxy nanocomposites. Overall, the manuscript is well written and clear, and provides positive contributions to the field. This manuscript contains some new and useful information to be published in Materials. However before accepted for publication, the following improvement should be made.
- Although the language expression is clear, there still exist some minor grammatical, syntax or word usage errors in the manuscript. The English language should be carefully corrected.
- The result presentation and discussion could be more extensive, especially the interfacial bonding of TiO2 and epoxy resin. Before it can be published, some complementarities should be included.
- The FT-IR spectrum of pure TiO2 particles should be provided.
- After reading the whole manuscript, it gives the overall impression that lacking of evidence or data to support “… good interfacial bonding of TiO2 in the epoxy resin results in improved mechanical properties with the addition of nanoparticles.”, “These PNCs exhibit excellent mechanical, physical, and chemical properties; however, these properties considerably depend on the interfacial bond strength between the polymer matrix and reinforcement particles.
- The stress-strain curves of TiO2–epoxy nanocomposites should be provided.
Author Response

(The authors gave the same response as above.)

Round 2
Reviewer 1 Report
I regret to say that I did not find significant improvements with respect to the previous version. Moreover the answers are NOT well-related to my comments.
Author Response

(The authors gave the same response as above.)

Reviewer 3 Report
The authors have properly addressed all my questions. I recommend it for publication as is.
Author Response
Dear reviewer,
First of all, thank you for your valuable comments concerning our manuscript entitled " Effect of Particle Size on the Mechanical Properties of TiO2–Epoxy Nanocomposites " by Young Min Choi, Seon Ae Hwangbo, Tae Geol Lee and Young Bog Ham* for publication in Micromachines journal.
Your feedback and comments have been all valuable and very helpful for revising and improving our paper. We have studied comments carefully and have made detailed and cautious English language corrections in addition to improving the quality of graphics used in manuscript.
Thank you very much for your help.
Kind regards,
Dr. Young Min Choi, Dr. Seon Ae Hwangbo, Dr. Tae Geol Lee and Dr. Young Bog Ham
Round 3
Reviewer 1 Report
I regret to say that I did not find significant improvements with respect to the previous version.